

# Enhancing intrusion detection performance using explainable ensemble deep learning

Chiheb Eddine Ben Ncir[1], Mohamed Aymen Ben HajKacem[2] and Mohammed Alattas[1]

[1] MIS Department, College of Business, University of Jeddah, Jeddah, Jeddah, Saudi Arabia
[2] LARODEC Lab, ISG Tunis, University of Tunis, Le Bardo, Tunis, Tunisia

## ABSTRACT

Given the exponential growth of available data in large networks, the need for an accurate and explainable intrusion detection system has become of high necessity to effectively discover attacks in such networks. To deal with this challenge, we propose a two-phase Explainable Ensemble deep learning-based method (EED) for intrusion detection. In the first phase, a new ensemble intrusion detection model using three one-dimensional long short-term memory networks (LSTM) is designed for an accurate attack identification. The outputs of three classifiers are aggregated using a meta-learner algorithm resulting in refined and improved results. In the second phase, interpretability and explainability of EED outputs are enhanced by leveraging the capabilities of SHape Additive exPplanations (SHAP). Factors contributing to the identification and classification of attacks are highlighted which allows security experts to understand and interpret the attack behavior and then implement effective response strategies to improve the network security. Experiments conducted on real datasets have shown the effectiveness of EED compared to conventional intrusion detection methods in terms of both accuracy and explainability. The EED method exhibits high accuracy in accurately identifying and classifying attacks while providing transparency and interpretability.

## INTRODUCTION

The rapid growth of technology has led to the emergence and development of various systems and networks that have transformed multiple aspects of our lives, including communication, business operations, education, healthcare, and entertainment. These systems usually contain a lot of critical data related to our daily lives, financial transactions, or any other important information. Consequently, they become prime targets for cyber-attacks *via* networks. Many hacker entities try to violate these systems by compromising communication networks. The 2020 NTSC (National Technology Security Coalition) security report highlights the alarming rise in cyber-attacks with approximately 620 million in accounts being compromised by hackers in 2019. The number of attacks has continued to increase in recent years, especially after the COVID-19 pandemic and the

Corresponding author
Chiheb Eddine Ben Ncir,
cbenncir@uj.edu.sa

widespread adoption of online systems and networks. The need for robust network security measures has become more critical. In this context, Intrusion Detection Systems (IDS) have emerged to secure computer systems and networks from dangerous unauthorized access to data and systems. IDS have evolved over the years and have been adapted to the changing types and forms of security threats. Initially developed in the 1980s (*Liao et al., 2013*), IDS emerged as a response to the growing need for defense against unauthorized access and malicious activities within computer systems. At that time, IDS was introduced as an additional dimension to security technologies, such as firewalls, able to monitor system traffic and detect potential intrusions. After that, with the growth of the internet and network environments, IDS has expanded to monitor network traffic and detect suspicious patterns or anomalies in such networks.

More recent IDS systems have been proposed by integrating artificial intelligence techniques, such as machine learning and deep learning, to effectively learn and identify patterns of normal and malicious behavior (*Ahmad et al., 2021*; *Khraisat et al., 2019*). Machine learning-based IDS have been proposed to automatically monitor user activities, system behavior, and network traffic aiming to identify deviations from expected patterns. However, a key challenge in applying machine learning models to detect and classify attacks in network traffic is the need for pre-configured input features. This limitation has been addressed through the utilization of deep learning-based models which automatically generate different features from traffic data and subsequently detect and classify attacks. Deep learning models have shown superior performance in terms of accuracy for attack detection compared to traditional machine learning models (*Lansky et al., 2021*).

Despite the efficiency of deep learning models for intrusion detection, they suffer from a lack of interpretability that is required to explain the obtained classification. The automatic detection operates as a "black-box" system that only gives a final classification of the network attacks without providing explanations of such classification and attack detection. Explanations may help security experts to transparently and effectively determine correct attacks and implement the right strategies to protect the system. Furthermore, existing approaches are usually based on a single deep neural network such as multi-layer perceptron (MLP) (*Shettar et al., 2021*), long short-term memory (LSTM) (*Laghrissi et al., 2021*), convolutional neural network (CNN) (*Nguyen & Kim, 2020*), and gated recurrent units (GRU) (*Cao et al., 2022*). However, since the results of a single deep learning model can be inaccurate or biased, it becomes more interesting to build a final decision based on an ensemble of models using ensemble learning methods.

To deal with all the discussed issues, we propose an Explainable Ensemble deep learning (EED) method for intrusion detection systems. The EED method consists of two main phases: *data detector modeling* and *model explaining*. In the first phase, we propose an ensemble intrusion detection model using three LSTM structures to classify network traffic. The outcomes of three LSTM classifiers are aggregated by a machine learning algorithm to obtain more reliable results compared to models based on a single classifier. Our choice of LSTM-based classifiers is due to their ability to handle sequential data and capture temporal dependencies over time. This makes them well-suited for modeling the sequential nature of the network traffic and more effective in identifying attacks. In the

second phase, we propose to explain the built intrusion detection model by using explainable artificial intelligence (XAI) technique called SHape Additive exPlanations (SHAP) (*Lundberg & Lee, 2017*). Such explanations make the detection and classification of attacks highly transparent for security experts. The major contributions of this article are summarized as follows:

- Design of a new ensemble deep learning model which consists of an application of three LSTMs with different structures to learn various patterns of network traffic; and then the integration of results using a classifier as a meta-learner for the combination and final assignment of the correct label to each attack.
- Development of an interpretation model for the generation of explanations based on the SHAP method that provides local and global explanations for the detection and classification of attacks. These explanations can be helpful to security experts for a deep understanding of output results.
- Comparative analysis of the effectiveness of conventional machine learning and deep learning models with EED to show the effectiveness of ensemble learning model on intrusion detection domain.

The remainder of this article is organized as follows: "Related Works" discusses a survey of recent related works while "Background and Preliminaries" presents the LSTM model followed by the SHAP technique. Then, "Proposed Explainable Ensemble Deep Learning Method for Intrusion Detection" describes the proposed EED for intrusion detection. "Experiments and Results" presents conducted experiments and obtained empirical results. Finally, "Conclusion" summarises this work and discusses the future directions.

## RELATED WORKS

This section begins by presenting the recent intrusion detection methods based on machine learning. Then, gives the state of the art of application of deep learning methods for intrusion detection and finally describes explainable artificial intelligence techniques for intrusion detection.

### Applications of machine learning models for intrusion detection

Machine learning models were widely used for intrusion detection (*Ahmad et al., 2021*; *Liu & Lang, 2019*). Common machine learning models used to detect attacks include decision trees (DT), random forest (RF), support vector machines (SVM) and k-nearest neighbors (KNN). *Debicha et al. (2021)* evaluated the performance of different machine learning models for intrusion detection. The experimental results showed that random forest achieved the best accuracy compared to decision trees and SVM for binary classification of attacks. *Dina, Siddique & Manivannan (2022)* proposed a machine learning-based intrusion detection system to classify attacks. The experimental results showed that the decision tree model outperforms RF and SVM models for multi-class classification of attacks.

*Sathesh (2019)* proposed an intrusion detection method based on a combination of feature selection, clustering, and classification techniques. This method applied a fuzzy

rule-based system to analyze the features followed by the application of DT model to select important features. To reduce the computational complexity, data are clustered using K-means to minimize the number of data in the training dataset. The SVM classifier is then used to categorize the attacks on the network. *Soheily-Khah, Marteau & Béchet (2018)* proposed a hybrid intrusion detection method based on K-means and random forest algorithm. The obtained results showed a better accuracy of attack prediction using the clustering technique. *Sahu & Mehtre (2015)* developed several DT-based models for intrusion detection such as C4.5, ID3 and CART. These models employ various techniques for feature selection, pruning, and handling missing values to build an efficient DT model for classifying network traffic.

*Aburomman & Reaz (2016)* proposed a machine learning-based intrusion detection model. They formulate the intrusion detection problem using the SVM classifier. The authors also used a radial basis function (RBF) kernel to categorize attacks into predetermined classes. The obtained results confirmed the suitability of SVM for intrusion detection tasks given its high accuracy compared to conventional machine learning methods. *Li et al. (2014)* also proposed an intrusion detection method based on KNN model. In the first step, the k-means clustering algorithm is used to build a cluster center for each group. Then, KNN is applied as a classification technique to determine the nearest neighbors. The experimental results showed that their proposed KNN-based classifier performs better than SVM in terms of classification accuracy. *Azimjonov & Kim (2024)* presented a novel approach for designing a lightweight intrusion detection system specifically for IoT networks. They utilized SVM as the underlying machine learning algorithm and employed feature selection methods to enhance the efficiency of the system. Four feature selection techniques were applied namely Importance Coefficient, Forward, Backward, and Correlation Coefficient. These methods were used to identify the most significant and effective features for detecting IoT botnet attacks.

Although the application of machine learning models has shown good accuracy in the detection and classification of attacks, it requires affordable efforts for pre-processing and configuring input features in a real-world attack investigation. Conventional machine learning models need predefined input features given as inputs for machine learning models. This problem is solved by using deep learning models that automatically extract and learn features from any input data (*Koumakis, 2020*). This important characteristic explains the effective and wide use of deep learning models in intrusion detection systems.

## Applications of deep learning models for intrusion detection

Deep learning models have been well applied and implemented in intrusion detection systems. Several deep learning models for intrusion detection were proposed in the literature (*Liu & Lang, 2019*; *Ahmad et al., 2021*). *Laghrissi et al. (2021)* proposed a deep learning approach for intrusion detection using LSTM. The authors studied the performance of the model for binary and multi-class attack classification. The experimental results showed the suitability of LSTM to improve the classification accuracy of attacks compared to conventional machine learning methods. *Muhuri et al. (2020)* developed a method for intrusion detection to classify attacks by combining a genetic

algorithm (GA) with LSTM neural network. They found that LSTM classifier combined with an optimal feature set improves the accuracy of intrusion detection. The experimental results performed on NSL-KDD dataset showed that GA can largely improve the classification accuracy of LSTM for both binary and multi-class classification. Obtained results using LSTM classifier outperformed those obtained using SVM and RF.

*Zhang et al. (2023)* introduced a novel approach that combines bidirectional long short-term memory (BiLSTM) with an attention mechanism for network intrusion detection. The proposed method leverages the advantages of BiLSTM in capturing dependency relationships between features. Additionally, an attention mechanism is employed to analyze the network traffic classification generated by the BiLSTM model. Experimental results on a real dataset demonstrate that the proposed method achieves higher detection accuracy compared to existing intrusion detection methods. *Nguyen & Kim (2020)* proposed a deep intrusion detection method based on CNN model. The authors first partitioned the features into four subsets using fuzzy c-means clustering and then converted them into grayscale format data. Subsequently, the CNN model was utilized for identifying attacks. Experimental results on the NSL-KDD dataset showed that the proposed CNN model achieves good accuracy compared to both machine learning and deep learning methods.

*Halbouni et al. (2022)* introduced a deep learning-based method for intrusion detection systems by combining various types of deep learning models such as CNN and LSTM. The authors also employed a feature selection technique to build the feature set. Experimental results on the NSL-KDD dataset demonstrated that the LSTM model achieved higher accuracy compared to the CNN model. *Cao et al. (2022)* proposed an intrusion detection approach utilizing a recurrent neural network based on GRU. The authors developed an efficient system that classifies network flow instances as normal or attack. A Pearson correlation feature selection algorithm was utilized to optimize the computational complexity. Experimental results on the CICIDS2018 dataset indicated that the GRU-based RNN outperformed existing deep learning models.

*Shettar et al. (2021)* also presented a multi-layer perceptron (MLP) model for classifying attacks. Experimental results on the NSL-KDD dataset demonstrated that the MLP model achieved higher accuracy compared to traditional machine learning models. *Jeyanthi & Indrani (2023)* proposed an intrusion detection system for healthcare applications using a deep learning model with custom features. The method incorporated a recurrent neural network (RNN) and a BiLSTM algorithm to detect and classify intrusion attacks. Custom features were extracted from incoming data streams and used to train the deep learning models. Experimental results showed the effectiveness of the proposed system in detecting and mitigating security threats in a healthcare environment.

Despite the high accuracy of deep-learning-based intrusion detection systems, they often operate as a "black-box" system that only gives a final classification of attacks without providing explanations for the reasoning behind the classification. Explanations may help security experts to transparently and effectively determine the correct attacks and implement the right strategies to protect the network.

## Explainable intrusion detection systems

Several works studied the explainability of intrusion detection systems (*Neupane et al., 2022*) through using explainable artificial intelligence methods such as SHAP (*Lundberg & Lee, 2017*) and Local Interpretable Model-agnostic Explanations (LIME) (*Ribeiro, Singh & Guestrin, 2016*). *Wang et al. (2020)* proposed a framework that enhances the explainability of intrusion detection systems. Their method leverages SHAP to generate both local and global interpretations. The local interpretation focuses on explaining the reasoning behind a specific decision using a single data record, whereas the global interpretation utilizes the entire dataset to provide insights into the overall structure of the model.

*Younisse, Ahmad & Abu Al-Haija (2022)* also proposed an explainable intrusion detection system. They combined deep neural networks and the interpretability of the model predictions. The proposed system utilized SHAP to provide both local and global explanations. *Keshk et al. (2023)* proposed an explainable intrusion detection system in IoT networks. They developed a deep-learning intrusion detection approach based on LSTM and generated explanations using several techniques such as SHAP, Permutation Feature Importance, Individual Conditional Expectation, and Partial Dependence Plot. The proposed system achieved high accuracy and high interpretability of output results compared to other IDS systems.

*Khan et al. (2021)* introduced an explainable auto-encoder-based detection framework for identifying attacks in IoT networks. The framework utilizes CNN and LSTM models to detect both known and zero-day attacks. The data are initially processed by an auto-encoder-based LSTM to extract temporal features, followed by the CNN model for generating predictions. The LIME is employed to provide further explanations regarding the detected attacks. Also, in the work of *Barnard, Marchetti & DaSilva (2022)*, an explainable intrusion detection method is proposed which integrates eXtreme Gradient Boosting (xgboost) as a classifier and SHAP as an explainable method. Despite the high detection accuracy, the proposed method has a high computational complexity due to the complexity of the explanation step. To reduce the computational cost, the authors utilized the PDP plots to eliminate features that cannot be adequately explained while maintaining nearly the same predictive accuracy.

*Bashaiwth, Binsalleeh & AsSadhan (2023)* proposed an explainable intrusion detection method that utilizes the LSTM model as a classifier. The authors combined both SHAP and LIME to enhance explainability. The combined XAI methods provide insights into the model's decision-making process and help identify the intrinsic features necessary to differentiate attacks. The proposed model offers a deeper understanding of the factors contributing to the classification of attacks. *Sharma et al. (2024)* also introduced a deep neural network approach for intrusion detection that utilized a filter-based approach to reduce the dimensionality and emphasize the most important aspects. This approach selects the most relevant features and limits the number of inputs to the model. Furthermore, the authors applied a set of XAI techniques to explain the attack detection. Furthermore, *Oseni et al. (2022)* introduced an explainable deep learning-based intrusion detection framework to enhance the transparency of deep learning-based IDS in IoT

**Table 1 Comparison between recent intrusion detection methods.**

| Reference | Year | ML | DL | XAI | Algorithm |
|---|---|---|---|---|---|
| *Soheily-Khah, Marteau & Béchet (2018)* | 2018 | √ | × | × | RF |
| *Sathesh (2019)* | 2019 | √ | × | × | SVM |
| *Muhuri et al. (2020)* | 2020 | × | √ | × | LSTM |
| *Wang et al. (2020)* | 2020 | × | √ | √ | SHAP |
| *Nguyen & Kim (2020)* | 2020 | × | √ | × | CNN |
| *Debicha et al. (2021)* | 2021 | √ | × | × | DT, RF, SVM |
| *Sahu & Mehtre (2015)* | 2021 | √ | × | × | DT |
| *Laghrissi et al. (2021)* | 2021 | × | √ | × | LSTM |
| *Shettar et al. (2021)* | 2021 | × | √ | × | MLP |
| *Khan et al. (2021)* | 2021 | × | √ | √ | CNN, LSTM, Autoencoder |
| *Dina, Siddique & Manivannan (2022)* | 2022 | √ | × | × | DT, RF, KNN |
| *Halbouni et al. (2022)* | 2022 | × | √ | × | CNN, LSTM |
| *Cao et al. (2022)* | 2022 | × | √ | × | GRU |
| *Younisse, Ahmad & Abu Al-Haija (2022)* | 2022 | × | √ | √ | SHAP |
| *Barnard, Marchetti & DaSilva (2022)* | 2022 | × | √ | √ | XG-boost, SHAP |
| *Oseni et al. (2022)* | 2022 | × | √ | √ | CNN, SHAP |
| *Zhang et al. (2023)* | 2023 | × | √ | × | BILSTM |
| *Jeyanthi & Indrani (2023)* | 2023 | × | √ | × | BILSTM |
| *Keshk et al. (2023)* | 2023 | × | √ | √ | LSTM, SHAP |
| *Bashaiwth, Binsalleeh & AsSadhan (2023)* | 2023 | × | √ | √ | LSTM, SHAP |
| *Azimjonov & Kim (2024)* | 2024 | √ | × | × | SVM |
| *Sharma et al. (2024)* | 2024 | × | √ | √ | DNN, SHAP, LIME |
| Proposed EED | – | × | √ | √ | Ensemble LSTM, SHAP |

networks. The framework incorporates the SHAP method to interpret the decisions made by the deep learning-based IDS. The authors provide insights and explanations regarding the reasoning behind the IDS's predictions which allows for a better understanding of the underlying factors contributing to the classification of intrusions. An overview of existing explainable approaches for intrusion detection systems is given in Table 1. All the described studies focused on enhancing the interpretability of intrusion detection systems using various methods such as SHAP, LIME, and ensemble models. While these approaches have shown promising results in providing explanations for detected attacks, they are usually based on a single deep learning model such as CNN, RNN, and LSTM. We will explore in the next sections the use of explainable techniques to interpret the results of ensemble deep-learning-based models in intrusion detection applications.

## BACKGROUND AND PRELIMINARIES

In this section, we first introduce the underlying mathematical model of LSTM and then present the main objectives and capabilities of the explainable artificial intelligence method SHAP.

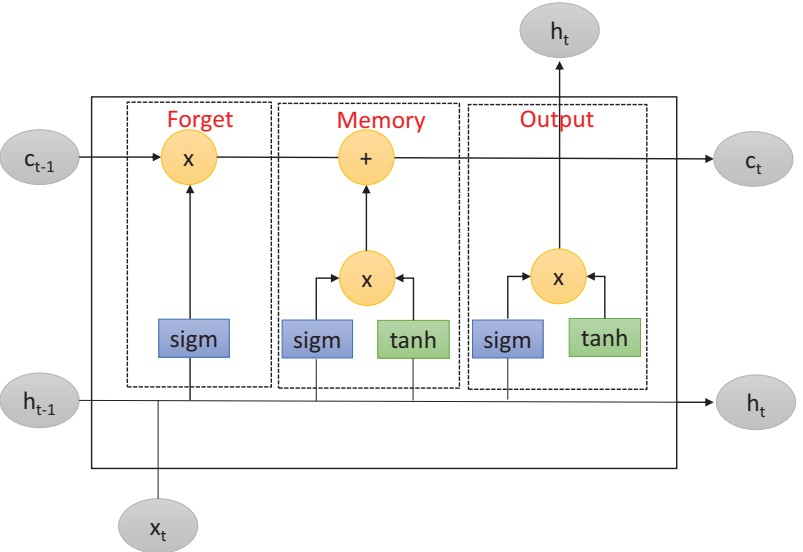

**Figure 1 Long short-term memory structure.**

## LSTM

LSTM is an improved version of RNN (*Hochreiter & Schmidhuber, 1997*) that solves the problem of long-term dependencies within RNN-based networks. This problem consists of the inability of RNNs to retain information from previous input data over longer sequences to improve the prediction. To overcome this limitation, LSTM includes a memory cell to learn which information will be retained or forgotten. As described in Fig. 1, the memory cell is controlled by three primary gates: the forget gate, the memory gate, and the output gate. These gates enable LSTM to effectively manage and regulate the flow of information within the network.

First, the forget gate determines which information from the previous cell state should be discarded. It takes as inputs the previous input layer $x_{t-1}$, the previous hidden layer $h_{t-1}$, and the previous cell state $c_{t-1}$. The forget gate applies the *sigmoid* function to these inputs, producing a real number between 0 and 1. This number represents the forget gate's decision regarding the amount of information from the previous cell state that should be forgotten. Mathematically, the forget gate operation can be described as follows:

$$f_t = sigm(W_f \cdot [h_{t-1}, x_{t-1}] + b_f) \tag{1}$$

where $f_t$ is the forget gate output at time-step $t$, *sigm* is the *sigmoid* function, $W_f$ is the weight matrix, $[h_{t-1}, x_{t-1}]$ is the concatenation of the previous hidden layer and the previous input layer, and $b_f$ is the bias term.

After that, in a second step, the memory gate determines which new information should be stored in the current cell state. It consists of two components: the input gate and the new candidate values. The input gate decides which values should be updated while the new candidate values are potential new values that could be added to the cell state. The input gate and candidate values are computed using the *sigmoid* and hyperbolic *tanh*

activation functions, respectively. Mathematically, the memory gate operations can be described as:

$$i_t = sigm(W_i \cdot [h_{t-1}, x_{t-1}] + b_i) \qquad (2)$$
$$\tilde{C}_t = \tanh(W_C \cdot [h_{t-1}, x_{t-1}] + b_C) \qquad (3)$$

where $i_t$ and $\tilde{C}_t$ are respectively the input gate output and the candidate value at time-step $t$, $W_i$ and $W_C$, are the weight matrices, $[h_{t-1}, x_{t-1}]$ is the concatenation result of the previous hidden layer and the previous input layer, and $b_i$ and $b_C$ are the bias terms.

Finally, the output gate regulates the output information from the LSTM cell. It takes as inputs the current input layer $x_t$, the previous hidden layer $h_{t-1}$, and the current cell state $C_t$. Similarly to the forget gate, the output gate uses the *sigmoid* function to determine which parts of the cell state should be returned as output. Additionally, the output gate applies the *tanh* function to the cell state to generate values between $-1$ and 1, which are then multiplied by the output of the *sigmoid* function. Mathematically, the output gate operation can be described as:

$$o_t = sigm(W_o \cdot [h_{t-1}, x_t] + b_o) \qquad (4)$$
$$h_t = o_t \cdot \tanh(C_t) \qquad (5)$$

where $o_t$ and $h_t$ are respectively the output gate and the current hidden layers outputs at time-step $t$, $h_t$, is the current hidden layer output at time step $t$, $W_o$ is the weight matrix, $[h_{t-1}, x_t]$ is the concatenation of the previous hidden layer and the current input layer, $b_o$ is the bias term, and $C_t$ is the current cell state.

We note here that other activation functions, such as rectified linear unit (ReLU) and Softmax, can be used as activation functions for the LSTM model. Our choice of *sigmoid* and hyperbolic *tanh* is based on their common use in RNN-based networks (*Hochreiter & Schmidhuber, 1997*; *Zhang et al., 2023*) and their successful applications in malware and intrusion detection contexts (*Muhuri et al., 2020*; *Nguyen & Kim, 2020*). Although *sigmoid* and hyperbolic *tanh* activation functions have certain limitations, such as being computationally consuming compared to other functions like ReLU and having a lower convergence speed during training, their widespread use within the LSTM model can be attributed to several reasons. Firstly, both functions are non-linear and capable of capturing complex patterns and relationships in the data. Secondly, the gradients of these functions can be easily computed, unlike other functions such as softmax, which allows to effectively optimize the network parameters during the learning process.

## SHAP

The SHAP technique was introduced by *Lundberg & Lee (2017)* to explain predictions built using supervised learning methods. It is based on the idea of Shapley value from game theory which quantifies the contribution of each feature to build the final predictions. The Shapley value measures the difference in prediction when a specific feature is included or excluded. It calculates the marginal contribution of each feature by considering all possible

combinations of features and their respective Shapley values. Suppose we have the following:

- $\mathscr{F}$ a set of features in a dataset
- $|\mathscr{F}|$ the total number of features.
- $A$ a subset of features used in the prediction model.
- $|A|$ the number of features in the subset $A$.
- $\Delta_i(A, x)$ is the difference between the prediction generated when including the feature $i$ and all the other possible predictions generated without including the feature $i$.

The Shapley value for a particular feature $i$, denoted as $SV_i(x)$, is calculated as the sum over all subsets $A$ that include the feature $i$ as follows:

$$SV_i(x) = \sum_{A \subseteq \mathscr{F} \setminus i} \frac{|A|! \times (|\mathscr{F}| - |A| - 1)!}{|\mathscr{F}|!} \times \Delta_i(A, x). \tag{6}$$

The calculated Shapley value $SV_i(x)$ provides insights into the marginal contribution of feature $i$ to the prediction of a single data record $x$. The importance of feature $i$ in the determination of the class label of the data record $x$ is quantified by considering all possible subsets of features with and without including that feature in the learning model. The contribution $SV_i(x)$ can be either positive or negative. A positive value indicates that feature $i$ positively influences the prediction, whereas a negative value suggests a negative influence.

The calculated $SV_i(x)$ only gives an interpretation of the importance of feature $i$ in determining the class label of the data record $x$ (*Qaffas et al., 2023a*). However, to build a global overview of the importance of each feature across all data records, global Shapley values can be calculated (*Qaffas et al., 2023b*) by summing local Shapley values for all data records $x \in X$ as in the following:

$$\overline{SV_i(x)} = \frac{\sum\limits_{x \in X} |SV_i(x)|}{|X|} \tag{7}$$

where $|X|$ is the number of data records in the dataset and $|SV_i(x)|$ is the absolute value of the local Shapley value for the data record $x$ on feature $i$. Global Shapley values provide a comprehensive understanding of features' importance and their variation across the whole predictive model enabling better interpretation of the learning model.

## PROPOSED EXPLAINABLE ENSEMBLE DEEP LEARNING METHOD FOR INTRUSION DETECTION

To simultaneously address the accuracy and explainability challenges of intrusion detection models, we propose a new design of an explainable ensemble deep learning method called EED. As shown in Fig. 2, the proposed EED method consists of two main phases: *data detector modeling* and *model explaining*.

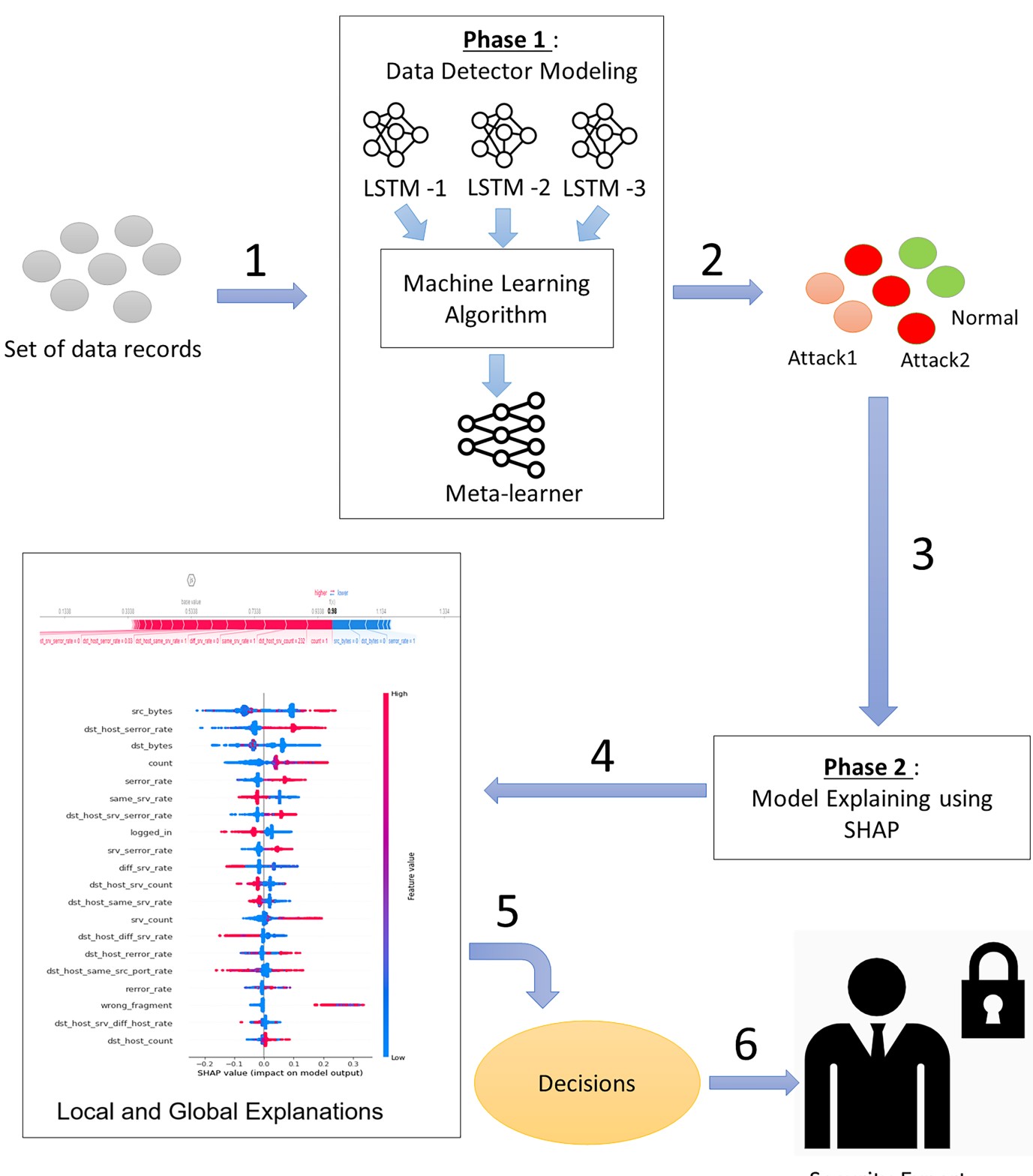

**Figure 2** **The main phases of the proposed explainable ensemble deep learning method for intrusion detection.** Phase 1: Data detector modeling and phase 2: Model explaining.

In the data detector modeling phase, we introduce an ensemble intrusion detection model that utilizes three LSTM structures to analyze and classify network traffic. The outcome of three classifiers is aggregated by a machine learning algorithm to obtain more reliable results than a single classifier. The inclusion of multiple classifiers allows for diversity in the modeling approach and helps reduce individual biases. The combination of three classifiers will give a good trade-off between accuracy and computational efficiency. Using fewer than three classifiers may limit diversity and robustness, whereas using more than three may lead to increased computational costs.

In the second phase (model explaining), we propose to explain the built intrusion detection model by exploiting XAI SHAP method capabilities. The transparency and the easy interpretation of the intrusion detection model are almost as important as the classification accuracy. For this reason, we design an intelligent SHAP-based process that offers explanations at both the global and local levels. These explanations provide insights into the overall structure of the intrusion detection results and also offer explanations specific to the classification of network traffic at the input level.

## Phase 1: Data detector modeling

The data detector modeling phase involves an ensemble deep learning model utilizing three LSTM structures. This phase is divided into two main stages: (1) the *base-learner* stage, where three distinct LSTM neural networks are employed, and (2) the *meta-learner* stage, which focuses on applying a machine learning algorithm to effectively aggregate the different results.

### LSTM model training

Deep learning models have shown superior accuracy compared to traditional machine learning models (*Lansky et al., 2021*). Specifically, LSTM-based models offer several advantages including ease of training and satisfactory performance even with limited computational resources. These characteristics make them highly suitable for addressing the challenges associated with intrusion detection (*Zhang et al., 2023*). We leverage the advantages of LSTM networks by proposing a design that includes three distinct LSTM configurations as base learners. The outputs of these base learners will be then aggregated using a meta-learner to build improved final results. The use of multiple LSTM base learners enables to explore different representations and variations in the data and increases the ability to capture complex relationships.

Initially, a simple LSTM classifier with a single LSTM layer is developed. Then, additional LSTM layers are added by using two key techniques: batch normalization and dropout. Batch normalization is applied to normalize the activation within each mini-batch to guarantee stable and consistent learning across the network (*Bjorck et al., 2018*). On the other hand, the dropout is applied to reduce network complexity by randomly deactivating neurons during training. This allows for preventing over-reliance on specific connections and improving generalization (*Baldi & Sadowski, 2013*). Furthermore, we introduce a flattened layer before the fully connected dense layer. This flattening operation reshapes the input data to ensure compatibility with the subsequent layers. The output of

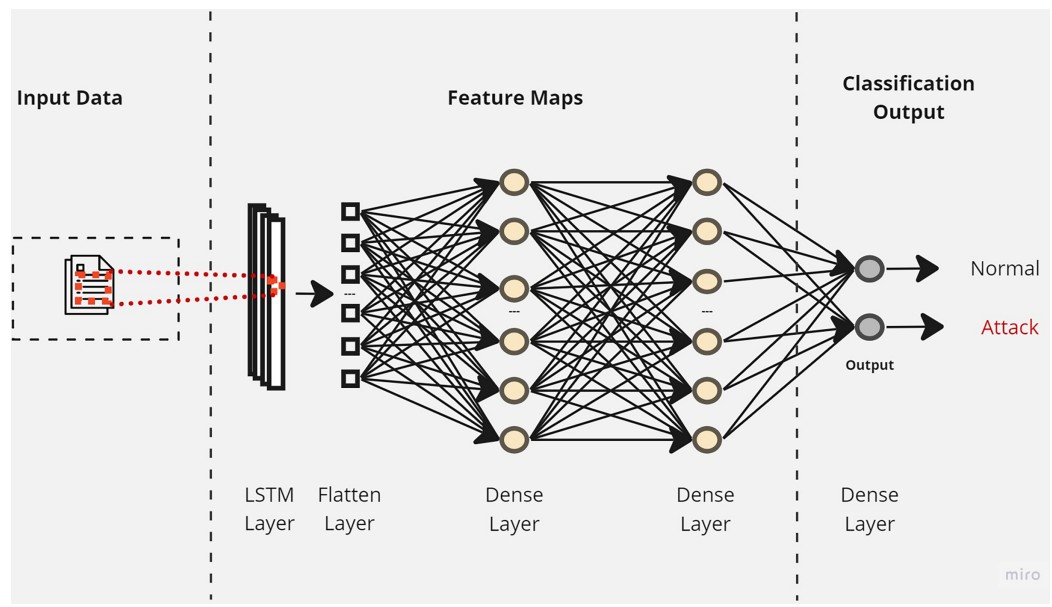
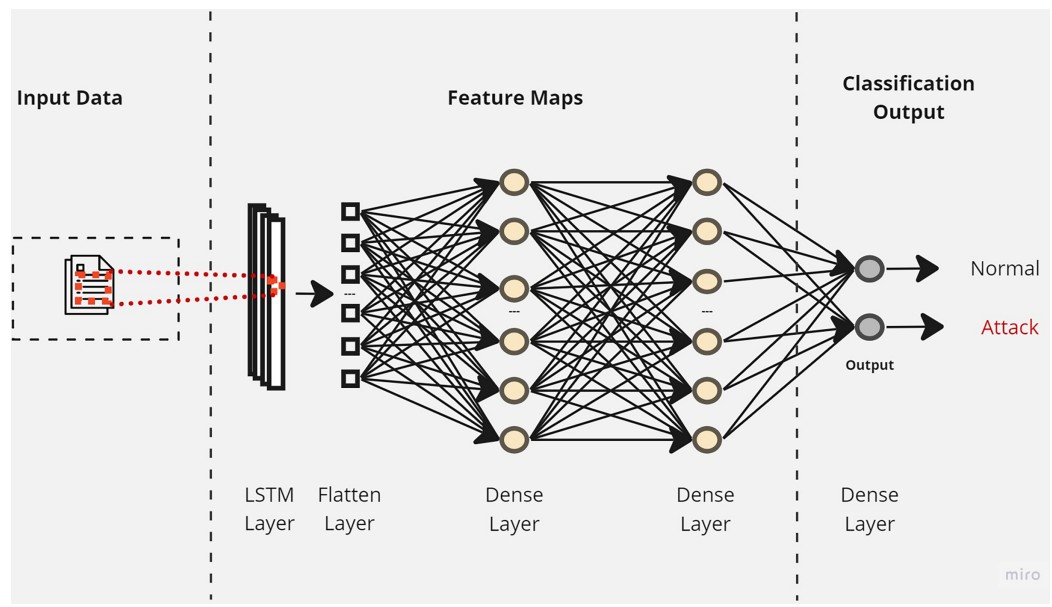

**Figure 3 The layers structure of the proposed LSTM-1.**

the last layer is considered as the final returned predictions. The proposed layers of the LSTM classifiers are illustrated in Figs. 3–5. These visual representations provide a clear overview of the architectural configuration for each classifier and highlight the arrangement and connectivity of the LSTM layers.

Concerning the activation of the output dense layer, we use the Softmax (*Liu et al., 2016*) function which can be defined as follows:

$$\sigma(x)_i = \frac{e^{x_i}}{\sum_{j=1}^{C} e^{x_j}} \tag{8}$$

where $x$ represents the input vector and $C$ denotes the total number of classes. The Softmax function plays a crucial role in transforming the raw scores into a normalized probability distribution across multiple classes. It is important to note that the Softmax function ensures a valid probability distribution given that all probabilities across all classes sum up to 1.

Concerning the loss function, we opt for the Categorical Cross-Entropy (CCE) (*Rusiecki, 2019*) due to its effectiveness in handling multi-class classification tasks. The CCE loss function is well-suited for LSTM models and has the purpose of quantifying the dissimilarity between the predicted probability distribution and the true class label distribution. CCE is mathematically defined as:

$$CCE = -\frac{1}{N}\sum_{i=1}^{N}\sum_{c=1}^{C}(p_{ic}\log(y_{ic})) \tag{9}$$

where $p_{ic}$ is a binary indicator function that indicates whether the data record $i$ belongs to category $c$ and $y_{ic}$ represents the predicted probability distribution for the data record $i$

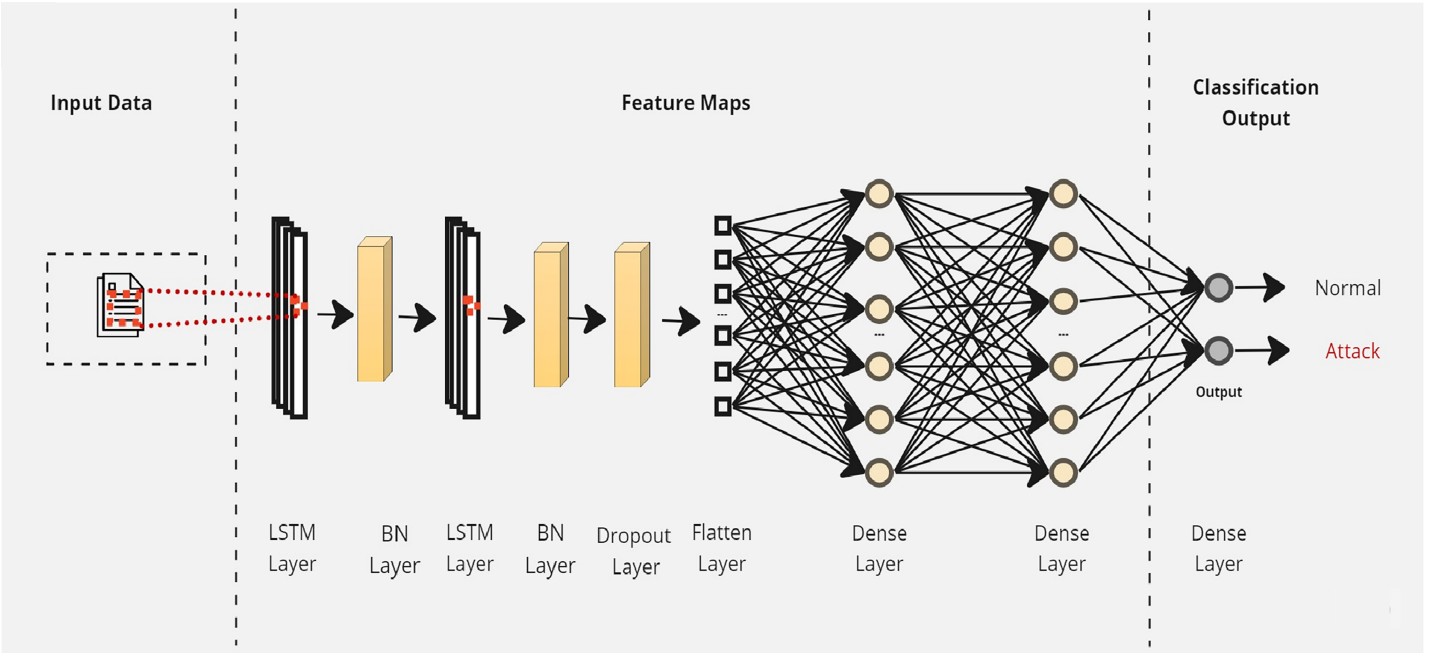

**Figure 4 The layers structure of the proposed LSTM-2.**

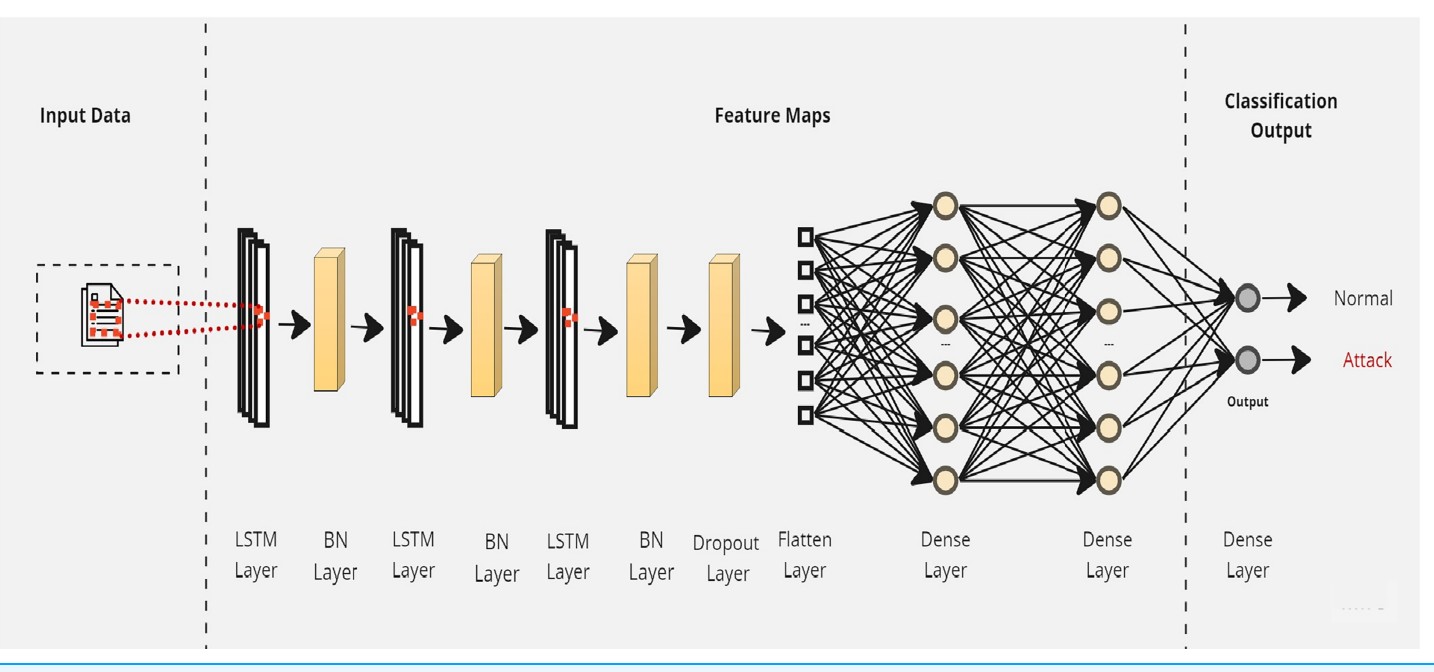

**Figure 5 The layers structure of the proposed LSTM-3.**

**Table 2 LSTM classifiers' architectures.**

| Parameter | LSTM-1 | LSTM-2 | LSTM-3 |
|---|---|---|---|
| No. of LSTM layers | 1 | 2 | 3 |
| No. of filters | 32 | 64 | 128 |
| Batch normalization | – | √ | √ |
| Dropout rate | – | 0.1 | 0.1 |
| Network optimizer | Adam | Adam | Adam |
| Kernel size | 3 | 5 | 7 |
| Batch size | 32 | 32 | 32 |
| Epochs | 50 | 100 | 100 |

belonging to class $c$. By minimizing the CCE loss, the model is trained to assign higher probabilities to correct classes and lower probabilities to incorrect ones.

Each LSTM model is trained using a $k$-fold cross-validation approach, where $k$ represents the number of folds. The training data is split into $k$ folds: $D = \{D_1, D_2, \ldots, D_k\}$. Each fold $D_i$ is used as a validation set and the remaining $(k-1)$ folds are used as a training set for each LSTM model. Once the models are trained, predictions from each LSTM model for the data records in fold $D_i$ are obtained. These predictions are denoted as $c1_i$, $c2_i$, and $c3_i$, representing the predicted class labels for LSTM1, LSTM2, and LSTM3, respectively. Given the crucial role of LSTM parameters in building effective learning models, we propose to explore different values for each parameter. We used the GridSearch algorithm (*Gozzoli, 2018*) to optimize all the LSTM parameters given its ease of implementation and its effective computational complexity. Selected parameter values for the three LSTM classifier configurations are reported in Table 2. Each classifier is designed with a unique combination of parameter values.

*Meta-learner model training*

Once the base learners are trained individually, the next step aims to apply a meta-learner to aggregate the results from each base classifier. The meta-learner is a machine learning algorithm that takes the output predictions of the base learners as input and generates the final predictions. To consolidate the predictions from each LSTM model across all folds, we combine them into a matrix with $n$ rows and $3k$ columns. This matrix, denoted as $M$, has the following structure: $M = [c1_1, c2_1, c3_1, \ldots, c1_k, c2_k, c3_k]$. Each column in $M$ represents the predictions from one LSTM model for a particular fold, and each row corresponds to a data record. A selected machine learning algorithm is trained using $M$ as the input data and the true class labels as the target variable. Several machine learning algorithms can be used at this step such as random forest, support vector machines, or K-nearest neighbor. By learning from the combined predictions of the LSTM models, the algorithm can effectively build the final predictions. To illustrate the overall process of the

---

| **Algorithm 1** The main steps of the proposed ensemble deep learning method. |
|---|
| **Input**: $D$: Traffic Data |
| **Output**: $P$: Final classification of traffic data $X$ |
| —- **LSTM Model Training** —- |
| 1. Split the training data into $k$ folds: $D = \{D_1, D_2, ..., D_k\}$. |
| 2. **For each** fold $D_i(i = 1 \quad to \quad i <= k)$ |
|     **For each** $LSTM_l$ model |
|         train $LSTM_l$ on $(k - 1)$ folds on $(D/D_i)$ |
|         Build the predictions from each $LSTM_l$ model for the data records in fold $D_i$ |
| 3. Combine the predictions of the 3 LSTM models on data records in fold $D_i$ such that $M = [M|[c1_i, c2_i, c3_i]]$ |
| —- **Meta-learner Model Training** —- |
| 4. Train the selected machine learning algorithm on $M = [c1_1, c2_1, c3_1, ..., c1_k, c2_k, c3_k]$ |
| 5. Use the meta-learner-trained model to build the final classification on new traffic data $X$ |
| 6. Return $P$ the final attack classification on traffic data $D$ |

whole proposed ensemble deep learning method, we provide in Algorithm 1 a step-by-step description of both LSTM model training and meta-learner model training stages.

## Phase 2: Model explaining

This phase aims to provide explanations for the classes generated in the previous phase. These explanations will allow cybersecurity experts to better understand and interpret the resulting classification of attacks at both local (individual traffic data) and global (attack class) levels.

Concerning local explanations, it focuses on explaining the predictions made for individual data traffic records and the reasons behind assigning them to specific attack class. To achieve this, we compute the Shapley value for each data record and each feature. The calculated Shapley values quantify the contribution of each attack feature, for each traffic record, towards the final prediction. These local explanations support security experts in comprehending the reasons behind assigning a connection to a particular normal or attack class. Furthermore, experts can analyze the positive and negative contributions of each feature for each connection, providing insights into the impact of individual features on the predicted class labels.

Let us consider a small traffic dataset with three features: "Source IP", "Destination IP", and "Protocol". We will focus on explaining the predictions made for individual data traffic records using local Shapley values. Suppose we have the following three data records:

- Record 1: Source IP: '192.168.0.1', Destination IP: '10.0.0.1', Protocol: 'TCP'
- Record 2: Source IP: '192.168.0.2', Destination IP: '10.0.0.2', Protocol: 'UDP'
- Record 3: Source IP: '192.168.0.3', Destination IP: '10.0.0.3', Protocol: 'ICMP'

For each record and each feature, we will calculate the Shapley value with respect to the assigned class (normal or attack). Let us assume the assigned class for each record is as follows:

- Record 1: Assigned Class → Normal
- Record 2: Assigned Class → Attack
- Record 3: Assigned Class → Normal

To compute the Shapley values, we analyze the contribution of each feature for each record towards the final prediction. Let us assume the Shapley values are calculated as follows:

- Record 1:

  - Shapley Value (Source IP) = 0.1
  - Shapley Value (Destination IP) = −0.2
  - Shapley Value (Protocol) = 0.3

- Record 2:

  - Shapley Value (Source IP) = −0.4
  - Shapley Value (Destination IP) = 0.5
  - Shapley Value (Protocol) = 0.2

- Record 3:

  - Shapley Value (Source IP) = 0.2
  - Shapley Value (Destination IP) = −0.3
  - Shapley Value (Protocol) = −0.1

These Shapley values quantify the contribution of each feature for each record. Positive values indicate a positive contribution towards the assigned class, whereas negative values indicate a negative contribution. Based on these local explanations, security experts can better understand the reasons behind assigning each connection to a specific normal or attack class. They can analyze the positive and negative contributions of each feature for each connection and gain insights into the impact of individual features on the predicted class label. For example, in Record 1, the "Source IP" and "Protocol" features have a positive contribution towards the assigned class while the "Destination IP" feature has a negative contribution.

To provide security experts with summarized class interpretations, we generate further explanations at the class level. These explanations allow identifying the most critical features when building each attack class and then offer valuable insights into the workings of the learning model. To determine the importance of each feature, we calculate the average of the absolute Shapley values locally for each class across all traffic records. The calculated global shapley values are typically sorted in descending order to show the most

important ones for each class. These important attack features provide insights into the proposed prediction model and its ability to distinguish between different attack classes which can help in better understanding the automatic recommendations built by the intrusion detection system.

Let us consider the small example of three records described above, global shapley values for each class can be calculated as follows:

- Assigned Class → Normal:

    – Global Shapley Value (Source IP) $= |0.1 + 0.2|/2 = 0.15$
    – Global Shapley Value (Destination IP) $= |(-0.2) + (-0.3)|/2 = 0.25$
    – Global Shapley Value (Protocol) $= |0.3 + (-0.1)|/2 = 0.1$

- Assigned Class → Attack:

    – Global Shapley Value (Source IP) $= |-0.4|/1 = 0.4$
    – Global Shapley Value (Destination IP) $= |0.5|/1 = 0.5$
    – Global Shapley Value (Protocol) $= |0.2|/1 = 0.2$

These global Shapley values represent the average absolute Shapley values per feature for each assigned class. These global Shapley values provide insights into the overall impact of each feature on the predictions for each class. They can help in understanding the relative importance of features for differentiating between attack classes. For example, for the 'Normal' class, the most important feature is 'Destination IP', followed by the 'Source IP' and 'Protocol' and also for the second class, "Attack", the most important feature is 'Destination IP', followed by the 'Source IP' and 'Protocol'.

## EXPERIMENTS AND RESULTS

### Evaluation methodology: datasets description and evaluation measures

Empirical experiments were conducted on a Dell Inspiron machine with an Intel[R] Core[TM] i7-1165G7 processor running at 2.80 GHz and 8.00 GB of RAM. All methods were implemented within Python 3.9.7 by using TensorFlow, Pandas, and Keras libraries and evaluated on two datasets, NSL-KDD (https://github.com/Mamcose/NSL-KDD-Network-Intrusion-Detection/tree/master) and UNSW-NB15 (https://research.unsw.edu.au/projects/unsw-nb15-dataset), which are specifically designed for intrusion detection systems. The first dataset, NSL-KDD, is an enhanced version of the widely used KDD-CUP99 dataset (https://kdd.ics.uci.edu/databases/kddcup99/kddcup99.html). Several improvements were made to ensure its suitability for robust intrusion detection analysis. Firstly, redundant and duplicate records were eliminated to prevent biases and ensure a balanced representation of data. Secondly, the distribution of records across different attack categories in the NSL-KDD dataset was adjusted to address the imbalances in the

original KDD-CUP99 dataset. The number of records in each category was adjusted inversely proportional to the percentage of records in the KDD-CUP99 dataset. This adjustment helps to create a more realistic and representative dataset for training and testing intrusion detection models. Each data record in the dataset comprises a total of 41 features, including three symbolic, seven binary, and 31 numerical features. Detailed descriptions of these features can be found in *Dhanabal & Shantharajah (2015)* where authors provide valuable insights into the dataset composition and characteristics. The NSL-KDD dataset classifies network traffic into normal traffic and four types of attacks, namely denial of service (DOS), probe (PROB), remote to local (R2L), and user to root (U2R).

Concerning the second dataset, UNSW-NB15, was created by *Moustafa & Slay (2015)* to address the need for a realistic and representative dataset for evaluating intrusion detection models. Unlike NSL-KDD dataset, UNSW-NB15 contains new attack and standard network traffic data. Each data record in the dataset is described by 47 features, including three symbolic, two binary, and 42 numerical features. Detailed descriptions of these features can be found in *Moustafa & Slay (2015)* where authors provide valuable insights into the dataset description. The UNSW-NB15 dataset classifies network traffic into normal traffic and nine types of attacks, namely Exploits, Generic, Reconnaissance, Analysis, Shellcode, DoS, Worms, Fuzzers and Backdoors.

To assess the effectiveness of the proposed method, we utilized four evaluation metrics: accuracy, precision, recall, and F1-score. These metrics provide a comprehensive analysis of the intrusion detection system's accuracy. Given that multiple classes are defined in each dataset, we considered the calculation of these measures for each class $i$ separately using the following formula:

$$Accuracy_i = \frac{TP + TN}{TP + TN + FP + FN} \tag{10}$$

$$Precision_i = \frac{TP}{TP + FP} \tag{11}$$

$$Recall_i = \frac{TP}{TP + FN} \tag{12}$$

$$F1-Score_i = \frac{2 * Precision * Recall}{Precision + Recall} \tag{13}$$

where a true positive (TP) indicates that the intrusion detection system accurately detects a specific attack that has occurred, a true negative (TN) indicates that the system correctly identifies a normal connection, a false positive (FP) indicates that the system erroneously detects an attack that did not occur and a false negative (FN) occurs when the intrusion detection system fails to detect an intrusion after a specific attack. $Accuracy_i$ represents the proportion of accurate predictions local to class $i$. $Precision_i$ measures the proportion of true positives compared to all correct predictions while $Recall_i$ measures the proportion of actual correctly predicted positive instances. The last measure, $F1-score_i$, is a harmonic mean of both $Precision_i$ and $Recall_i$. All these described measures are calculated for each

**Table 3 Evaluation of EED performance using different meta-learner algorithms.** The best obtained values are highlighted in bold.

| Dataset | Model | Accuracy (%) | Precision (%) | Recall (%) | F1-score (%) |
|---------|-------|--------------|---------------|------------|--------------|
| NSL-KDD | EED (DT) | 96.14 | 96.47 | 96.23 | 96.77 |
| | EED (RF) | **97.05** | **97.38** | **97.15** | **97.19** |
| | EED (K-NN) | 95.39 | 95.97 | 95.67 | 95.88 |
| | EED (SVM) | 95.14 | 95.19 | 95.22 | 95.38 |
| UNSW-NB15 | EED (DT) | 93.25 | 93.47 | 91.19 | 92.35 |
| | EED (RF) | **95.15** | **95.88** | **95.01** | **95.25** |
| | EED (K-NN) | 93.21 | 93.66 | 93.61 | 92.88 |
| | EED (SVM) | 95.18 | 95.68 | 95.71 | 95.49 |

class $i$ and then an average over all classes (macro-averaging) is computed as in the following:

$$Accuracy = \frac{1}{N}\sum_{i=1}^{N} Accuracy_i \tag{14}$$

$$Precision = \frac{1}{N}\sum_{i=1}^{N} Precision_i \tag{15}$$

$$Recall = \frac{1}{N}\sum_{i=1}^{N} Recall_i \tag{16}$$

$$F1-Score = \frac{1}{N}\sum_{i=1}^{N} F1-Score_i \tag{17}$$

where $N$ is the total number of classes. All of these measures provide a quantitative assessment of the performance of each evaluated method, where higher values indicate better performance in identifying network intrusions.

## Performance evaluation of ensemble learning

In this section, we present the evaluation of the performance of our proposed ensemble deep learning method. Firstly, we conducted experiments to assess the accuracy of the meta-learner algorithm used in EED. We evaluated the performance of EED by using various machine meta-learning algorithms, including DT, RF, K-NN, and SVM. Obtained results on NSL-KDD and UNSW-NB15 datasets are summarized in Table 3. The reported results indicate that using Random Forest as a meta-learner for the proposed EED method achieved the highest accuracy for both datasets. This is explained by its robustness and ability to handle various data types and structures. Consequently, the Random Forest will be considered as the default meta-learner algorithm for all the next presented experiments to assess EED performance.

Then, we evaluated the accuracy when varying the k-fold cross-validation parameter. The evaluation results are presented in Table 4. Reported results of NSL-KDD demonstrate

**Table 4 Evaluation of the performance of EED (RF) when changing k-fold cross-validation parameter.** The best obtained values are highlighted in bold.

| Dataset | Model | Accuracy (%) | Precision (%) | Recall (%) | F1-score (%) |
|---------|-------|--------------|---------------|------------|--------------|
| NSL-KDD | 2 | 96.14 | 96.47 | 96.23 | 96.77 |
| | 4 | 95.89 | 95.35 | 95.22 | 95.74 |
| | 6 | **96.99** | **96.01** | **96.39** | **97.01** |
| | 8 | **97.05** | **97.38** | **97.15** | **97.19** |
| | 10 | 96.59 | 96.39 | 96.99 | 96.31 |
| UNSW-NB15 | 2 | 93.14 | 93.47 | 93.23 | 93.77 |
| | 4 | 94.06 | 94.25 | 94.11 | 94.88 |
| | 6 | 94.99 | 94.28 | 94.18 | 94.88 |
| | 8 | **95.15** | **95.88** | **95.01** | **95.25** |
| | 10 | 93.98 | 93.19 | 93.57 | 93.18 |

**Table 5 Comparison of the accuracy of the proposed ensemble method with individual LSTM models.** The best obtained values are highlighted in bold.

| Dataset | Classifier | Accuracy (%) | Precision (%) | Recall (%) | F1-score (%) |
|---------|-----------|--------------|---------------|------------|--------------|
| NSL-KDD | LSTM-1 | 96.15 | 96.11 | 96.48 | 96.19 |
| | LSTM-2 | 96.04 | 96.15 | 96.34 | 96.08 |
| | LSTM-3 | 96.31 | 96.15 | 96.49 | 96.13 |
| | EED | **97.05** | **97.38** | **97.15** | **97.19** |
| UNSW-NB15 | LSTM-1 | 93.89 | 93.87 | 93.11 | 93.19 |
| | LSTM-2 | 94.28 | 94.87 | 94.27 | 94.18 |
| | LSTM-3 | 94.54 | 94.15 | 94.55 | 94.28 |
| | EED | **95.15** | **95.88** | **95.01** | **95.25** |

that EED achieved the highest accuracy when $k = 8$ and $k = 6$ with 96.99 and 97.05 respectively. However, the obtained results of UNSW-NB15 dataset clearly show that EED achieved the highest accuracy when $k = 8$ with 95.15. Consequently, we decided to continue evaluations with $k = 8$.

Next, we evaluated the accuracy for each LSTM classifier as outlined in "Phase 1: Data detector modeling". We individually assessed the performance of each LSTM classifier and compared it to the performance of the ensemble model. The comparative results on the NSL-KDD and UNSW-NB15 datasets are presented in Table 5. Reported results on both datasets demonstrate that our proposed ensemble learning model outperformed all individual LSTM classifiers in terms of all evaluation measures. These experiments confirm that combining the outcomes of different LSTM structures significantly enhances the overall model performance. The Random Forest algorithm as the meta-learner and the combination of diverse LSTM classifiers contribute to achieving superior performance compared to individual LSTM models.

**Table 6 Comparison of the accuracy of EED with existing intrusion detection methods.** The best obtained values are highlighted in bold.

| Dataset | Type | Model | Accuracy (%) | Precision (%) | Recall (%) | F1-score (%) |
|---------|------|-------|--------------|---------------|------------|--------------|
| NSL-KDD | Machine learning models | DT | 95.14 | 95.36 | 95.48 | 95.39 |
| | | RF | 96.59 | 96.57 | 96.78 | 96.66 |
| | | K-NN | 94.13 | 94.01 | 94.38 | 94.25 |
| | | SVM | 94.58 | 94.51 | 94.79 | 94.68 |
| | Deep learning models | MLP | 94.18 | 94.29 | 94.22 | 94.38 |
| | | LSTM | 96.15 | 96.11 | 96.48 | 96.19 |
| | | CNN | 95.77 | 95.47 | 95.68 | 95.39 |
| | | GRU | 96.57 | 96.51 | 96.98 | 96.82 |
| | | EED | **97.05** | **97.38** | **97.15** | **97.19** |
| UNSW-NB15 | Machine learning models | DT | 93.87 | 93.22 | 93.48 | 93.79 |
| | | RF | 94.59 | 94.11 | 94.18 | 94.02 |
| | | K-NN | 92.89 | 92.11 | 93.38 | 92.45 |
| | | SVM | 94.12 | 94.51 | 94.64 | 94.84 |
| | Deep learning models | MLP | 94.01 | 94.28 | 94.98 | 94.18 |
| | | LSTM | 93.88 | 93.28 | 93.18 | 93.27 |
| | | CNN | 93.48 | 93.48 | 93.24 | 93.84 |
| | | GRU | 94.54 | 94.28 | 94.11 | 94.57 |
| | | EED | **95.15** | **95.88** | **95.01** | **95.25** |

## Comparison between EED and existing intrusion detection methods

We assessed the accuracy of the proposed EED method compared to existing machine learning and deep learning-based intrusion detection methods. The comparative results are presented in Table 6. Reported results of NSL-KDD dataset demonstrate that EED achieved the highest accuracy among the compared methods. Our model attained an accuracy value of 97.05% surpassing all machine learning algorithms such as the decision tree-based method that achieved an accuracy of 95.17%, the K-nearest neighbor approach with 94.13% and the SVM method with 94.58%. Similarly, the accuracy achieved by our model on UNSW-NB15 dataset was 95.15% outperforming other machine learning algorithms, such as the decision tree-based method with 93.87% and the SVM method with 94.12%. These results can be explained by the effectiveness of the designed LSTM layers in building and extracting better input features from the dataset.

Furthermore, when compared to deep learning-based models, our EED model also demonstrated superior accuracy for both datasets. For example, the Multilayer Perceptron achieved an accuracy of 94.18%, the CNN achieved 95.77%, and GRU achieved 96.57% for NSL-KDD dataset. The higher accuracy of our EED model compared to deep-learning-based methods can be explained by the strength of ensemble learning to combine results of multiple deep models to enhance the overall performance. Therefore, the obtained results highlight the effectiveness and superiority of the proposed EED method compared to existing machine learning and deep learning-based intrusion detection methods. The

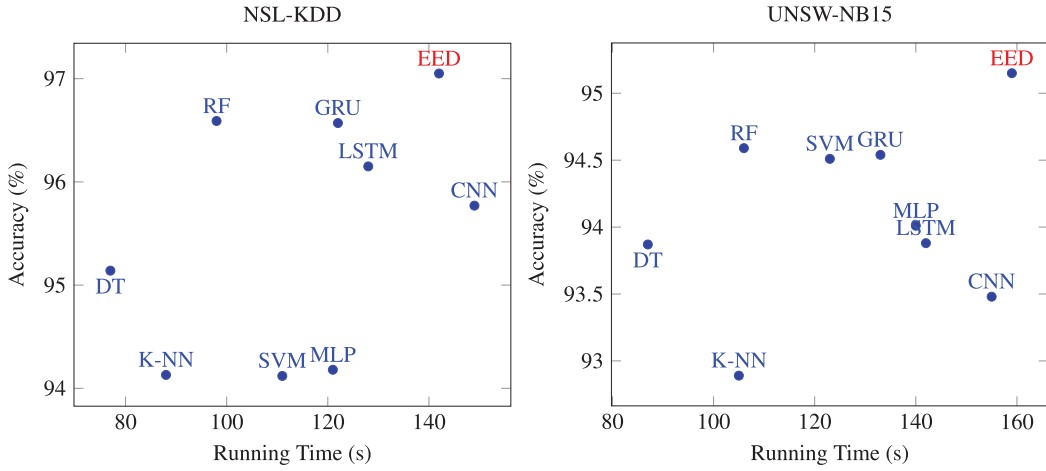

**Figure 6 Comparison of the running times of EED with existing methods.**

utilization of ensemble learning, along with the designed LSTM layers, has resulted in enhanced intrusion detection capabilities.

Then, we evaluated the running times of the proposed method compared to existing methods. The comparative results are presented in Fig. 6. Machine learning based method spends less time on each dataset, but its performance is not as good as other deep learning methods. Most of deep learning methods are time-consuming on the two datasets. Although the running time values of deep learning methods are high, their performance is excellent compared to machine learning. Our method showed a relatively comparable running times to deep learning-based methods while achieving superior accuracy. Thus, our proposed EED method is relatively time consuming with enhanced detection accuracy.

## Generating local and global explanations

To provide local explanations, we used the force plot which shows the contribution of each feature in the model's decision-making process. Figure 7A presents a local explanation for an average of 10 data records from NSL-KDD datasets, correctly classified by the model as a DOS attack. The feature *same_srv_rate* with a value of 0.01 significantly influences the decision to assign records to this class. A lower value of *same_srv_rate* increases the probability of assigning the record to the DoS attack class. Additionally, *diff_srv_rate* with a value of 0.08 and *count* with a value of 103 also contribute to the assignment of records to the DoS attack class. Figure 7B displays a local explanation for an average of 10 data records classified as a Probe (PROB) attack. The plot reveals that feature values such as *src_bytes* equal to 0, *dst_host_rerror_rate* equal to 0.25 and *dst_bytes* equal to 0 predominantly contribute to the decision of classifying a record as a PROB attack. It is important to note that *dst_host_same_srv* with a value of 0.52 decreases the probability of assigning a record to the PROB attack class. Concerning Figure 7C, it illustrates a local explanation for 10 data records correctly classified as R2L attack. The values of features such as *num_compromised = 2*, *hot = 3*, and *src_bytes = 2.42* significantly influence the

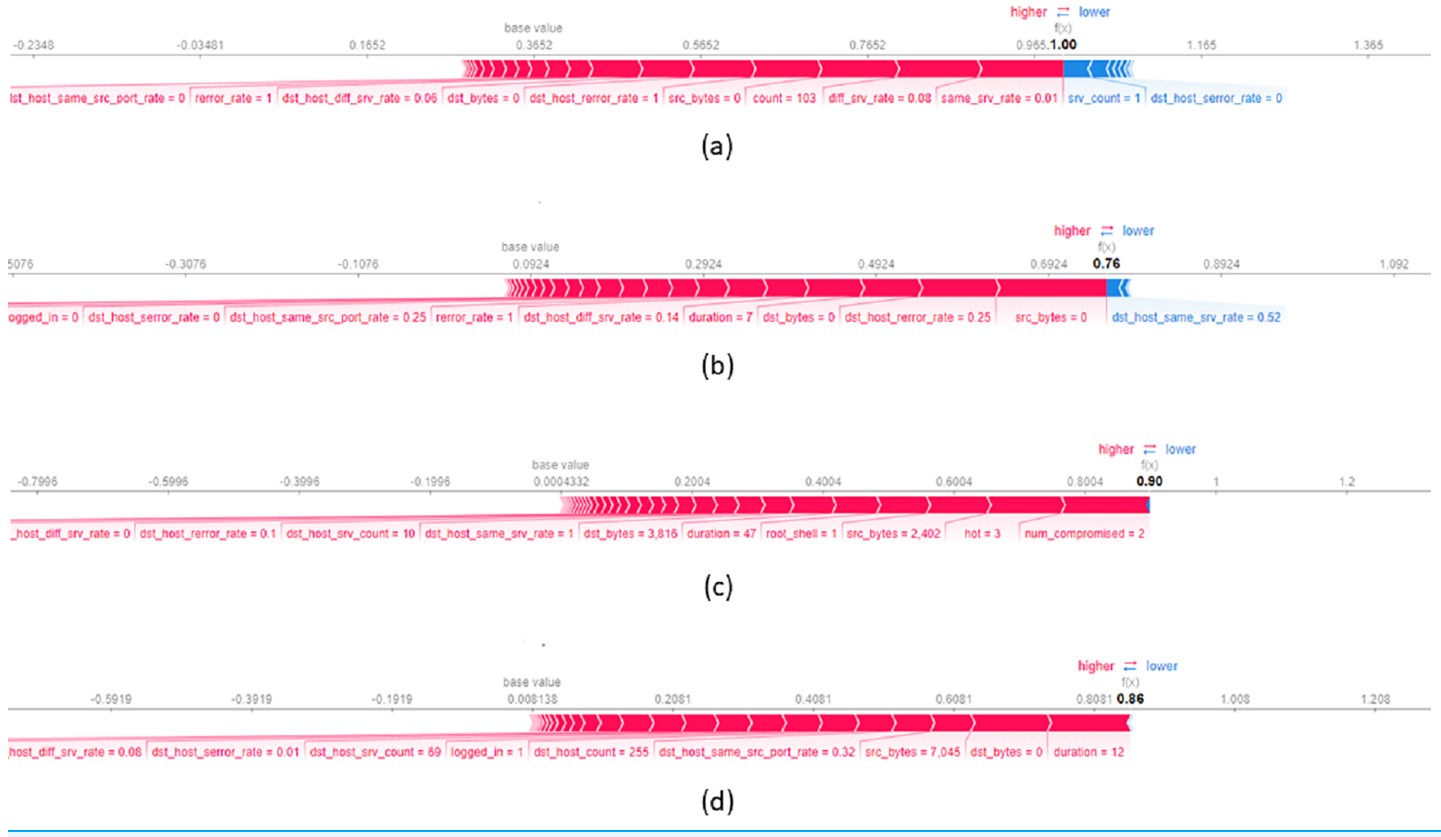

**Figure 7** Force plot of an average of 10 data records from NSL-KDD dataset, assigned respectively to the attack classes (A) DOS, (B) PROB, (C) R2L and (D) U2R.

decision to classify a record as an R2L attack. Higher values of these features largely increase the probability of classifying the record as an R2L attack. The last Fig. 7D presents a local explanation for 10 data records correctly assigned to the User to U2R attack class. The feature *duration* with a value of 12 positively contributes to the final classification indicating that a large duration value increases the likelihood of the connection being assigned to the U2R attack class. Moreover, *dst_bytes = 0* and *src_bytes = 7.045* also positively contribute to this decision.

Figure 8A presents a local explanation for an average of 10 data records from UNSW-NB15 dataset which are correctly classified by the model as a Generic attack. The feature *sbytes* with a value of 1.76 significantly influences the decision to assign records to this class. A higher value of *sttl* increases the probability of assigning the record to the Generic attack class. In addition, *smean* with a value of 638 and *ct_dst_src_ltm* with a value of 2 also contribute to the assignment of records to the Generic attack class. Concerning Fig. 8B, it shows a local explanation for 10 data records correctly classified as Analysis attack. The values of features such as *dbytes = 0.001*, *sttl = 62*, and *dmean = 439* significantly contribute the decision to classify a record as an Analysis attack. Higher values of these features largely increase the probability of assigning the record as an Analysis attack. Figure 8C

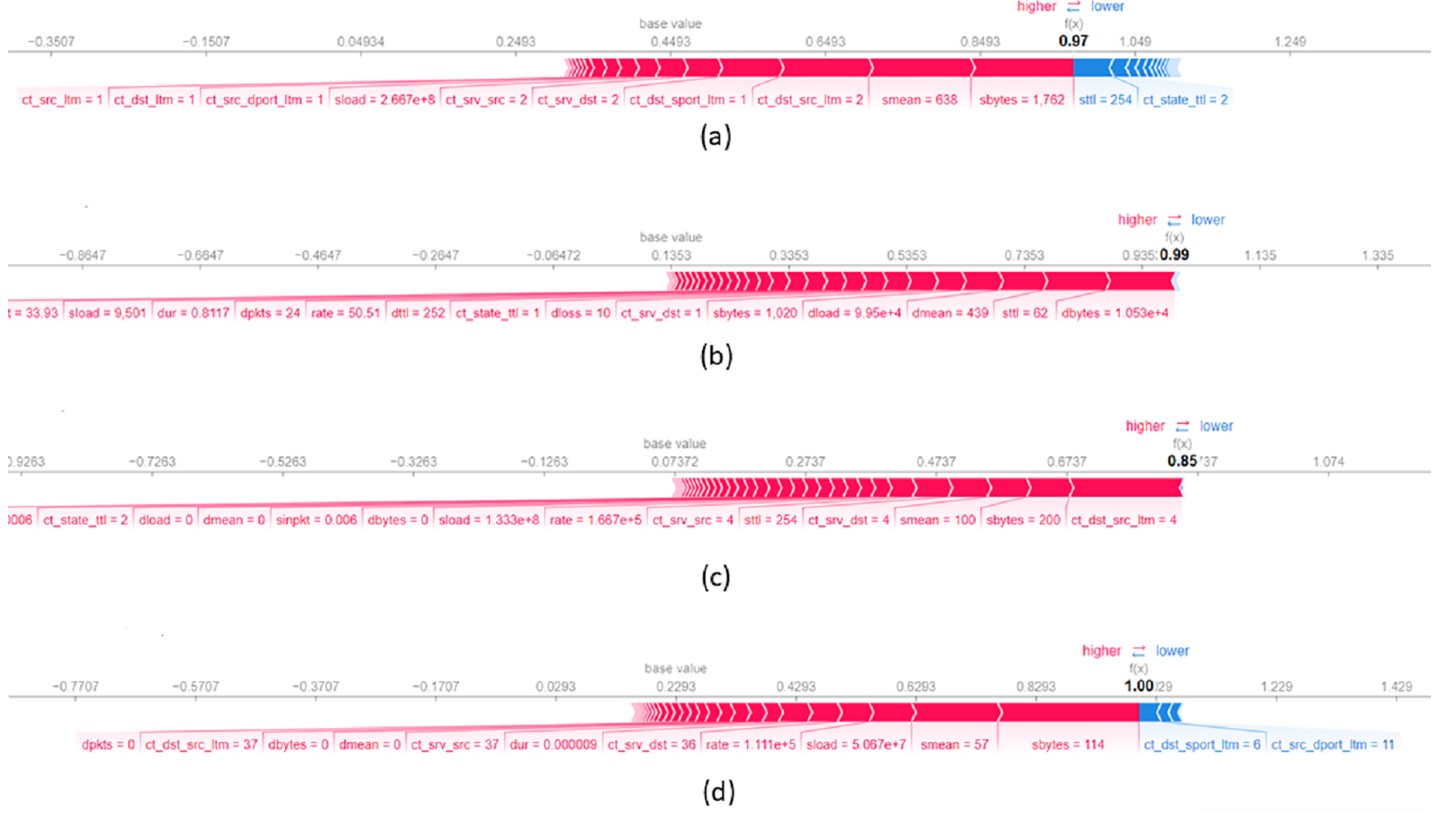

**Figure 8 Force plot of an average of 10 data records from UNSW-NB15 dataset, assigned respectively to the attack classes (A) Generic, (B) Analysis, (C) DoS and (D) Backdoors.**

illustrates a local explanation for an average of 10 data records classified as a DoS attack. This figure shows that feature values such as *ct_dst_src_ltm* equal to 4, *sbytes* equal to 200 and *seman* equal to 100 scientifically increase the probability of classifying a record as a DoS attack. Lastly, Fig. 8D presents a local explanation for 10 data records correctly assigned to the Backdoors attack class. The feature *sbytes* with a value of 114 positively contributes to the final classification indicating that a large value of this feature increases the likelihood of the data record being classified to the Backdoors attack class. In addition, a lower value of *ct_dst_sport_ltm* increases the probability of assigning the record to the Backdoors attack class.

In order to provide insights into the important features that were crucial in building attack classes, we build global explanations of EED results by using the SHAP feature-summary-plot as schematized in Figs. 9 and 10 for NSL-KDD and UNSW-NB15 datasets respectively. In these plots, each dot corresponds to a data record in the dataset. The vertical position of a dot represents a specific feature, while the horizontal position indicates the impact of that feature's value on the model's classes (local to each class individually). The color of each dot represents the value of the corresponding feature for that record in the dataset, with red indicating high values, purple indicating medium

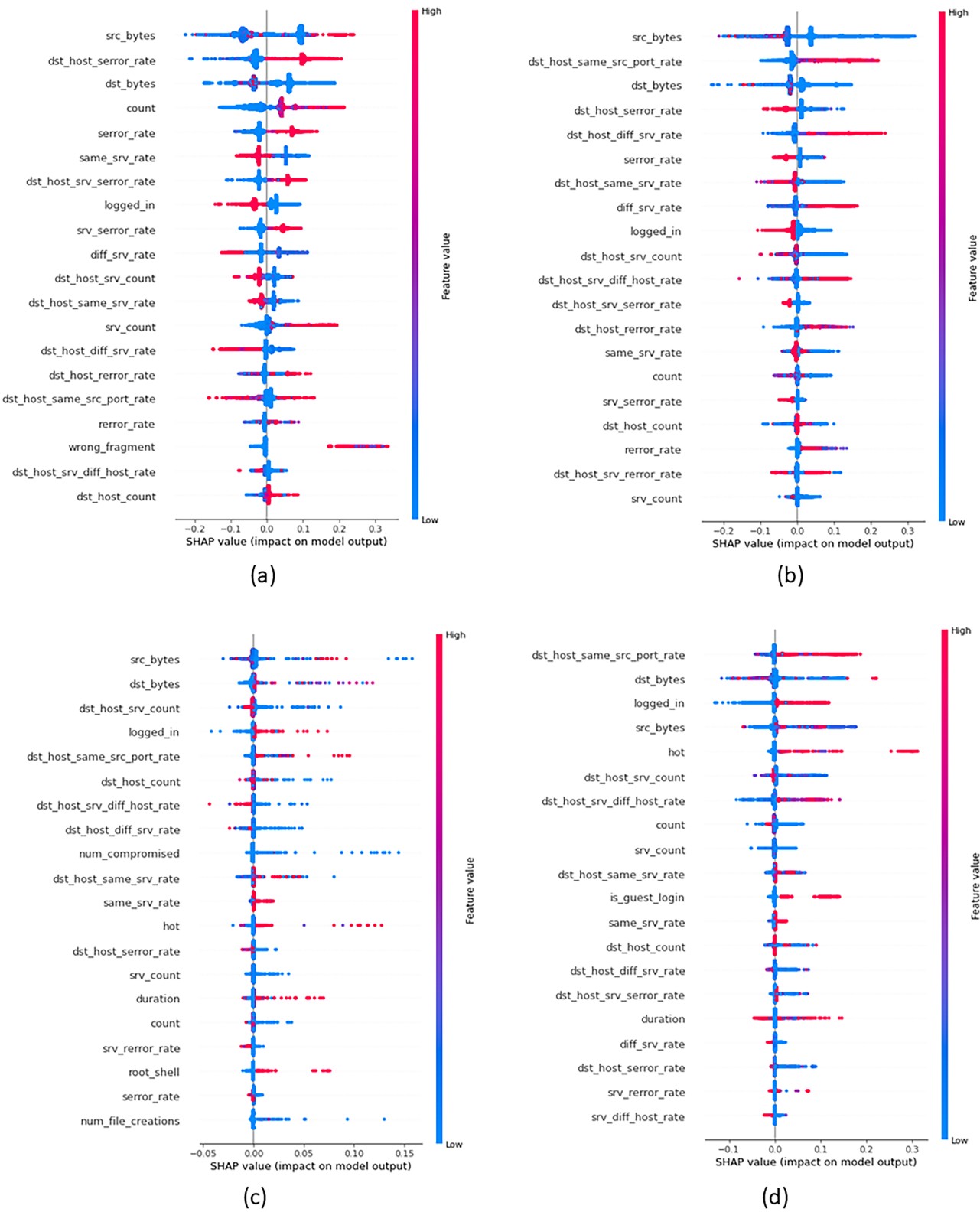

**Figure 9** Top 20 important features that were crucial in building each attack class from NSL-KDD dataset (A) DOS Class, (B) PROB class, (C) R2L class and (D) U2R class.

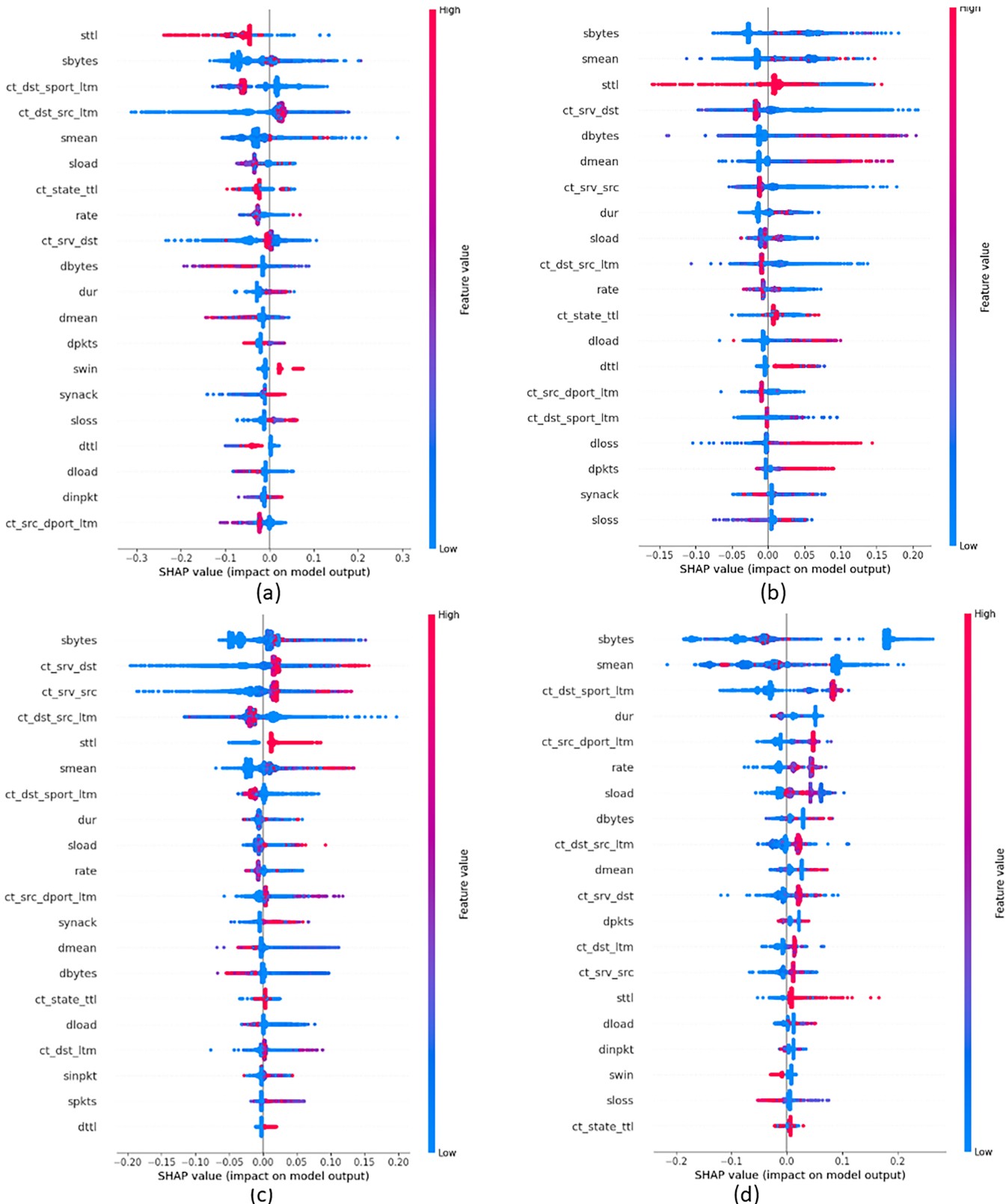

**Figure 10** Top 20 important features that were crucial in building each attack class from UNSW-NB15 dataset (A) Generic class, (B) Analysis class, (C) DOS class and (D) Backdoors class.

values, and blue indicating low values. Figure 9A focuses on the identification of DOS attacks. It shows that high values of the *src_bytes* feature increase the probability of a connection being classified as a DOS attack by 20% to 30%. Additionally, low values of the *dst_bytes* feature contribute to the identification of DOS attacks. Moving to the PROB attack class in Fig. 9B, explanations reveal that low values of the *src_bytes* feature increase the probability of a prediction being classified as a PROB attack by 30%. Moreover, large values of the *dst_host_same_srv* feature raise the probability of the connection being classified as a PROB attack by 10% to 30%.

Concerning Fig. 9C, it demonstrates the global explanation of the R2L attack class. It indicates that low values of the *dst_srv_count* feature increase the probability of a prediction being classified as an R2L attack, with a range of 0% to 30%. Also, large values of the *logged_in* feature increase the probability of the connection being classified as an R2L attack, ranging from 0% to 30%. Lastly, Fig. 9D provides the global explanation of the U2R attack class. It shows that low values of the *dst_srv_count* feature increase the probability of considering records as U2R attacks, ranging from 0% to 30%. Additionally, large values of both *dst_host_same_src* and *logged_in* features increase the probability of the connection being classified as a PROB attack, with a range of 0% to 20%.

Figure 10A displays the global explanation of Generic attack class. It shows that low values of the *sttl* feature increase the probability of a connection being classified as a DOS attack by 20% to 30%. Moreover, high values of the *smean* feature influence the classification of Generic attacks. However, reported results in Fig. 10B show that high values of the *smean* feature raise the probability of assigning a connection as an Analysis attack by 15%. In addition, low values of the *sbytes* feature increase the probability of the data record to be classified as an Analysis attack by 20%. Moving to Fig. 10C that gives explanations of the DoS attack class, reported results show that low values of *ct_srv_dst* feature contribute significantly to the classification of data records as a DoS attack, with a range of 0% to 30%. Similarly, large values of the *sbyte* feature increase the probability of the connection to be classified as a DoS attack with a range of 5% to 20%. Finally, Fig. 10D displays the global explanation of the Backdoors attack class. It indicates that low values of both *sbytes* and *smean* features increase the probability of the connection being classified as a Backdoors attack, with a range of 20% to 30%. Moreover, large values of the *ct_dst_sport_ltm* feature increase the probability of a connection being classified as Backdoors attacks, ranging from 10% to 20%.

## CONCLUSION

We proposed in this work an explainable ensemble deep learning method for an effective intrusion detection. The proposed method deals with the issues of accuracy and explainability while detecting attacks. It is based on ensemble deep learning and explainable artificial intelligence capabilities to build local and global explanations. It includes two phases: data detector modeling and model explaining. The first phase aims to identify attacks using three LSTMS classifiers. The results of three classifiers is aggregated by the random forest algorithm to obtain improved results compared to a single classifier.

The second phase is devoted to explain the model's predictions by generating local and global explanations using SHAP technique. Empirical experiments performed on real datasets have shown a significant improvement of both accuracy and explainability.

In this work, the obtained results heavily depend on the quality of the selected features. The initial feature selection greatly influences both the model outputs and the interpretability of the results. To enhance the identification and detection of attacks, it would be beneficial to incorporate a feature selection step that identifies most relevant features. This would help in improving the accuracy and explainability of the model.

In addition, integrating other XAI techniques, such as Local Rule-based Explanations (LORE), could further enhance the interpretability of the model. LORE can facilitate the generation of explanations in the form of simple rules making them intuitive and easily understandable, even for non-experts. Coupling LORE with SHAP would provide refined explanations where SHAP will focus on interpreting impacts of individual features while LORE will focus on generating global rule-based explanations. Furthermore, a potential future direction for the improvement of this work is to explore the impact of incorporating more than three classifiers in the ensemble intrusion detection model. By increasing the number of classifiers, we can further enhance the accuracy of the system. However, it is essential to evaluate the feasibility and computational costs associated with integrating a larger number of classifiers.

### Funding
This work was funded by the University of Jeddah, Jeddah, Saudi Arabia, under grant No. (UJ-24-DR-20621-1). The funders had no role in study design, data collection and analysis, decision to publish, or preparation of the manuscript.

### Grant Disclosures
The following grant information was disclosed by the authors:
University of Jeddah, Jeddah, Saudi Arabia: UJ-24-DR-20621-1.

### Competing Interests
The authors declare that they have no competing interests.

### Author Contributions
- Chiheb Eddine Ben Ncir conceived and designed the experiments, performed the experiments, analyzed the data, performed the computation work, authored or reviewed drafts of the article, and approved the final draft.
- Mohamed Aymen Ben HajKacem performed the experiments, analyzed the data, performed the computation work, prepared figures and/or tables, and approved the final draft.
- Mohammed Alattas conceived and designed the experiments, authored or reviewed drafts of the article, and approved the final draft.

## Data Availability
The code is available at GitHub and Zenodo:

- https://github.com/aymenhk1/EEDID

- aymenhk1, & Chihebncir. (2024). aymenhk1/EEDID: EEDID V1 (0.0.1.1). Zenodo. https://doi.org/10.5281/zenodo.13330555.

The NSL-KDD dataset is available at the Information Security Center of Excellence (INSCOE) at the University of New Brunswick (UNB) in Canada: https://github.com/Mamcose/NSL-KDD-Network-Intrusion-Detection/tree/master.

The UNSW-NB15 dataset is available at The Australian Centre for Cyber Security (ACCS) at the University of New South Wales (UNSW) in Canberra, Australia: https://research.unsw.edu.au/projects/unsw-nb15-dataset.

## Supplemental Information
Supplemental information for this article can be found online at http://dx.doi.org/10.7717/peerj-cs.2289#supplemental-information.

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
