# Peer review of "Enhancing intrusion detection performance using explainable ensemble deep learning"

_PeerJ Computer Science, doi:10.7717/peerj-cs.2289_

## Round 0.1 · original submission · Major Revisions

· Academic Editor

Major Revisions

We have completed 1st round review of your manuscript by 3 independent reviewers. They have suggested a major revision. One of the reviewers has also unloaded an annotated manuscript which you also need to consider during revision. You are required to revise and resubmit your manuscript. The revised manuscript will be subjected to a 2nd round review. Good luck.

**Language Note:** PeerJ staff have identified that the English language needs to be improved. When you prepare your next revision, please either (i) have a colleague who is proficient in English and familiar with the subject matter review your manuscript, or (ii) contact a professional editing service to review your manuscript. PeerJ can provide language editing services - you can contact us at [email protected] for pricing (be sure to provide your manuscript number and title). – PeerJ Staff

Reviewer 1 ·

Basic reporting

The introductory section provides a general overview of the Intrusion Detection, its growing popularity, and challenges related to security and privacy. However, it might benefit from a more detailed historical context and specific examples illustrating the evolution and current state of IDS. This would enhance the readers' understanding of the topic and its relevance. Several new research studies should be included in the references for the topic. Authors must enhance the topics included in the recently proposed models by several researchers.

Experimental design

The research objectives appears to be well-defined and relevant, focusing on addressing intrusion detection methods. The methods described, including the use of the famous dataset and the NSL-KDD dataset, seem appropriate for the research objectives. However, there is room for more explicit detailing of the procedures and protocols followed in the study to enable replication and assessment by other researchers.
Several questions like below should be answered by the authors to enhance the level of the article to the journal level as:
Why did the authors selected sigmoid and hyperbolic tangent tanh activation functions only?
Why did the authors used Shapley value identification and prediction method only?
Authors claim that the outcome of three classifiers is aggregated by a machine learning algorithm to obtain more reliable results than a single classifier. They are advised to add the necessity of the three classifiers. Precisely, why no more than 3 classifiers are used ? or why no less than 2 classifiers are used?
Why is the data detector modelling make use of LSTM only??
The manuscript lacks the answer to the questions above. Adding the reasons will add a lot of value to the article.
Figure 2 should be splitted up in two figures to make the phase 1 and phase 2 more explanatory. Authors should split the figure for clear understanding of the two phases.

Validity of the findings

The findings, including the proposed model’s high accuracy in intrusion detection, are promising. However, the article might benefit from a more rigorous statistical analysis of the results. Additionally, comparisons with existing models or systems in similar domains could provide a clearer understanding of the impact and novelty of the study.
Several questions should be answered as below:
How is the training data splitted in k folds? The value of k is determined by what factors and why?
Feature selection in the article remained ambiguous. It is a necessary addition to tell about the feature selection in the manuscript to reveal the need for the attack class.
Authors must integrate the feature selection methodology to ensure that the manuscript become more effective.

Additional comments

Various components like the intrusion detection, machine learning and neural network are discussed with technical details. However, a more comprehensive explanation of how these components interact and complement each other in the proposed system would be beneficial. Also, elaborating on the specific innovations or improvements each component brings to the IDS field would add value. The article covers data detection and intrusion detection as critical aspects of the system. Further clarification on how these processes are specifically tailored or optimized for the IDS context could enhance the article's relevance and applicability. Some references in the manuscript are not cited properly. It seems that the formatting of the article needs some modifications to make it acceptable as per the journal guidelines.

Reviewer 2 ·

Basic reporting

The topic chosen for the study is a very recent and popular topic. Especially in recent years, ensemble learning techniques have been preferred by many researchers. In this sense, it was a correct approach for researchers to aim to detect intrusion using a popular deep learning algorithm. However, the problems I will explain in the following sections do not comply with the currentness and professionalism of the study.

Experimental design

The general outline of the study is well prepared. The parts where they provide information about the literature review and the material seem sufficient. However, the parts I will mention below are too incomplete for the study;
1) The formulas of the performance metrics they give belong to binary classification, but it seems that they make multi label classification in experimental studies. In this case, there is a contradiction between the formulas and the operations performed.
2) No information is given about development environments. Which programming language was used? Information about what kind of operating system and what hardware they worked with should be clear.
3) How did they choose the parameters in Table 2? Is it based on existing studies in the literature or is it an intuitive approach?

Validity of the findings

1) The most important problem with the study is the dataset they used. Although KDD Cup is popular, it is now quite old. Why did they choose an old dataset when there were more up-to-date datasets? Did they evaluate current datasets or did they just use this old dataset?
2) Although the LSTM algorithm is a very successful deep learning algorithm, it is a time-consuming. When ensemble learning is involved, the learning time also increases. However, researchers have not provided clear information on this subject.
3) They should clearly explain why KDD Cup is used and why they do not use more recent datasets.
4) In Table 3 and Table 4, all metrics for RF and EED are exactly the same. I haven't seen an explanation as to why this might happen. In addition, while RF is a successful collective learning algorithm that is already accepted in the literature, a clear explanation is needed on why researchers should use their algorithm.

Additional comments

In the study, intrusion detection using ensemble model with deep learning is good. However, if the problems I mentioned are not clarified, paper will not be sufficient to be published in the journal.

·

Basic reporting

The authors propose a new enhancement of Deep Learning methods in intrusion detection. The new two-phase ensemble Deep Learning method, called EED (Explainable Ensemble Deep learning) addresses the need for accurate and explainable intrusion detection in networks. The proposed method is of high interest and lies very important perspectives in the field of Computer Science and its applications. The authors provide sufficient field Background for their research.

Experimental design

The proposed method consists of two phases. In the first phase, the authors propose an ensemble intrusion detection model using three Long Short-Term Memory (LSTM) models. The accuracy of attack identification is improved by aggregating the outputs of these deep learning classifiers with a meta-learner algorithm.

In the second phase, the authors focus on improving the interpretation and explanation of the detections tracked in Phase I. Based on the SHape Additive exPplanations (SHAP) capabilities, the authors highlighted the factors contributing to the identification and classification of attacks. These explanations provide a better understanding of detected attacks and allow to assist experts in developing effective response strategies to enhance network security.

While the paper makes significant contributions, some limitations need to be addressed to enhance the quality of the paper:

1- Regarding the choice of Categorical Cross-Entropy (CCE) for the LSTM models, the authors should provide a clear justification for this selection and provide any empirical evidence or prior work supporting its effectiveness in this context.

2- The description of existing explainable intrusion detection methods in Related Works section is insufficient. The authors must provide a critical analysis of the existing works listed in Table 1, highlighting their strengths and weaknesses, and explaining how their proposed method complements or improves upon these approaches. This allows for the reader to better understand the utility of explanations in intrusion detection systems.

3- The description of SHAP (SHapley Additive exPLanations) should be enhanced by including a brief explanation of SHAP plots or visualizations. Including such visualizations will strengthen the paper's explanation of the SHAP method.
4- The formula description for the LSTM model can be improved to enhance clarity and understanding. (Formula 1 to 5). The authors should provide a more detailed explanation of the different components and variables in the equations, ensuring that readers can follow the mathematical formulation without ambiguity.
5- Consistency in notation should be maintained throughout the paper. The authors should use the same notation for intervals, whether it is [h_t−1, x_t−1] or [0..1]. This will avoid confusion and enhance the overall readability of the paper.
6- In the conclusion section, the authors may provide further details on potential future works
7- Figure 2 can be improved by numbering the arrows to indicate the sequential steps. This numbering will provide a clear visual flow and help readers understand the sequential nature of the process.

Validity of the findings

The novelty of the method is assured. All underlying Data have been provided and explained. Empirical experiments are conducted on the NSL-KDD dataset to demonstrate the effectiveness of the proposed method in terms of accuracy and explainability. Conclusion are well stated and linked to the original research.

However, the paper should provide a justification for choosing SHAP (SHapley Additive exPLanations) over other explainable methods. The authors can discuss the specific advantages of SHAP in the context of intrusion detection. The authors may highlight why SHAP is the most suitable choice for the proposed method.

Additional comments

Well written article with a relevant applications of Deep Learning methods. I accept the article under the above mentionned minor revision.

---

## Round 0.2 · Minor Revisions

· Academic Editor

Minor Revisions

Please address the following concerns, and provide a reasonable response:

1) The title "Interpretable Ensemble Deep-Learning for Intrusion Detection: enhancing detection performance and explainability" has repetitive terms such as detection occurring twice, there is a dash (-) between deep and learning. Also the terms "Interpretable" and "explainability" are synonymous. The title needs to be modified. My suggestion would be "Enhancing intrusion detection performance using interpretable ensemble deep learning".
2) Some of the terms have been abbreviated several times. For instance, LSTM has been abbreviated in line no. 62, 90, 233. Proof read the entire manuscript for such redundancies. Check for SVM and other abbreviations and correct it.
3) In Table 1. Comparison between recent intrusion detection methods, Column "ML" and "DL", what does it mean? In my opinion, "DL" is a subfield of "ML". Further, many of the methods compared in this table do not belong of XAI.
4) Do the proposed method, "Ensemble LSTM", comes under XAI? If yes, it must be justified.

Reviewer 1 ·

Basic reporting

The authors have resolved all the queries raised during the review.
Significant changes are made, and the article is modified to a more significant extent.

Experimental design

The experimental analysis and design are improved to make it better.
The paper is now able to express the experimental analysis and results.

Validity of the findings

The outcomes are well-defined and are properly shaped for publication.

Additional comments

The complete manuscript is revised.
Proper corrections and improvements are reflected as per the Tracked Change Manuscript.
The author has significantly raised the level of work from the initial iteration.
The article will surely add value to the Journal Articles' level, and it is per the journal's guidelines.
The manuscript is acceptable in its present format.

·

Basic reporting

I thank the authors for considering all the comments and revising all the commented sections in their new manuscript. All questions and comments were explained and detailed very carefully. I accept the publication of this article in its new form.

Experimental design

I thank the authors for considering all the comments and revising all the commented sections in their new manuscript. All questions and comments were explained and detailed very carefully. I accept the publication of this article in its new form.

Validity of the findings

I thank the authors for considering all the comments and revising all the commented sections in their new manuscript. All questions and comments were explained and detailed very carefully. I accept the publication of this article in its new form.

Additional comments

no more comments. The article is very well written and the idea is very interesting.

---

## Round 0.3 · accepted · Accept

· Academic Editor

Accept

I am pleased to accept your paper for publication in PeerJ Computer Science. Your manuscript has undergone rigorous peer review, and I am delighted to say that it has been met with praise from our reviewers and editorial team. Your research makes a significant contribution to the field, and we believe it will be of great interest to our readership. On behalf of the editorial board, I extend our warmest congratulations to you.